# Molecular and Genomic Characterization of the *Pseudomonas syringae* Phylogroup 4: An Emerging Pathogen of *Arabidopsis thaliana* and *Nicotiana benthamiana*

**DOI:** 10.3390/microorganisms10040707

**Published:** 2022-03-24

**Authors:** Diego Zavala, Isabel Fuenzalida, María Victoria Gangas, Micaela Peppino Margutti, Claudia Bartoli, Fabrice Roux, Claudio Meneses, Ariel Herrera-Vásquez, Francisca Blanco-Herrera

**Affiliations:** 1Centro de Biotecnología Vegetal, Facultad de Ciencias de la Vida, Universidad Andres Bello, Santiago 8370146, Chile; diegoczt@gmail.com (D.Z.); ifuenzalida2009@gmail.com (I.F.); mariavictoria.gangas@gmail.com (M.V.G.); 2Center of Applied Ecology and Sustainability (CAPES), Departamento de Ecología, Pontificia Universidad Católica de Chile, Santiago 8320000, Chile; 3Millennium Science Initiative Program (ANID), Millennium Institute for Integrative Biology (iBio), Santiago, Chile; 4Millennium Science Initiative Program (ANID), Millennium Nucleus for the Development of Super Adaptable Plants (MN-SAP), Santiago, Chile; clamenes@gmail.com; 5Centro de Investigaciones en Química Biológica de Córdoba, CIQUIBIC, CONICET, Departamento de Química Biológica Ranwel Caputto, Facultad de Ciencias Químicas, Universidad Nacional de Córdoba, Córdoba X5000HUA, Argentina; mpeppino@unc.edu.ar; 6IGEPP, INRAE, Institut Agro, University Rennes, 35653 Le Rheu, France; claudia.bartoli-kautsky@inrae.fr; 7Laboratoire des Interactions Plantes-Microbes-Environnement (LIPME), INRAE, CNRS, Université de Toulouse, 31326 Castanet-Tolosan, France; fabrice.roux@inrae.fr; 8Departamento de Fruticultura y Enologiá, Facultad de Agronomiá e Ingenieriá Forestal, Pontificia Universidad Católica de Chile, Santiago 7820436, Chile; 9Departamento de Genética Molecular y Microbiologiá, Facultad de Ciencias Biológicas, Pontificia Universidad Católica de Chile, Santiago 8331150, Chile; 10Fondo de Desarrollo de Areas Prioritarias, Center for Genome Regulation, Santiago 8370415, Chile

**Keywords:** *Pseudomonas syringae*, *Arabidopsis thaliana*, disease emergence, pathogenic reservoirs, HopBJ1, virulence factor, T3SS

## Abstract

Environmental fluctuations such as increased temperature, water availability, and air CO_2_ concentration triggered by climate change influence plant disease dynamics by affecting hosts, pathogens, and their interactions. Here, we describe a newly discovered *Pseudomonas syringae* strain found in a natural population of *Arabidopsis thaliana* collected from the southwest of France. This strain, called *Psy* RAYR-BL, is highly virulent on natural Arabidopsis accessions, Arabidopsis model accession Columbia 0, and tobacco plants. Despite the severe disease phenotype caused by the *Psy* RAYR-BL strain, we identified a reduced repertoire of putative Type III virulence effectors by genomic sequencing compared to *P. syringae* pv *tomato* (*Pst*) DC3000. Furthermore, *hopBJ1_Psy_* is found exclusively on the *Psy* RAYR-BL genome but not in the *Pst* DC3000 genome. The plant expression of HopBJ1_Psy_ induces ROS accumulation and cell death. In addition, HopBJ1_Psy_ participates as a virulence factor in this plant-pathogen interaction, likely explaining the severity of the disease symptoms. This research describes the characterization of a newly discovered plant pathogen strain and possible virulence mechanisms underlying the infection process shaped by natural and changing environmental conditions.

## 1. Introduction

The disease triangle describes the variables that allow the development of an infectious process. This triangle includes the pathogen, the host, and the environmental conditions. The interaction between these three components conditions the plant’s resistance to the pathogen or disease development.

In this context, anthropogenic climate change generates detrimental environmental conditions such as increased atmospheric CO_2_, recurrent episodes of extreme temperatures, and drastic variations on water availability such as drought or increased precipitations [1]. These environmental changes modulate the plant defense response and affect the virulence mechanisms of pathogens [2]. Therefore, climate change can influence the outbreak of emerging plant diseases caused by changes in the pathogenesis of known pests, the increase in the incidence of infections in different geographical regions, or the colonization and infective success over new plant hosts [3].

Several outbreaks of plant pathogens belonging to the Pseudomonas genus have been described in the last decades, such as *P. savastanoi* pv. *mandevillae* affecting the ornamental plant Brazilian jasmine [4], the pathogen *P. fuscovaginae* that affects the Gramineae family producing significant loss on rice crops [5], or the *P. syringae* pv *actinidiae* that led to the devasting loss in the kiwifruit orchards worldwide during the last years [6,7].

Notably, *P. syringae* is one of the most studied bacterial plant pathogens isolated from a broad spectrum of hosts [8]. The most studied *P. syringae* bacterial models have been isolated mainly from tomato (*P. syringae* pv. *tomato*) [9] and cauliflower (*P. syringae* pv. *maculicola*) [10]. Despite being isolated from a crop species, the interaction between *P. syringae* pv. *tomato (Pst)* DC3000 and the model plant *A. thaliana* has been extensively used to establish the basis of our current knowledge of plant defense response and, on the other side, of the virulence mechanisms of bacterial infections. A primary virulence mechanism that favors the infection of *P. syringae* is its capacity to suppress the host immunity [11]. Thus, inducing an antioxidant imbalance increasing the ROS accumulation acts as a damaging agent, producing oxidation of biomolecules affecting the cell metabolism, manifesting as disease and tissue cell death [12].

The Type III secretion system (T3SS) is a syringe-like structure required to affect the plant defense response. When expressed, the T3SS forms a channel that extends from the bacterium to the plant cell membrane going through the cell wall [13]. Knock-out *P. syringae* mutants on the components that participate in the T3SS assembly fail to induce disease symptoms. For example, the *P. syringae* hrcC^−^ strain, lacking the *hrcC* gene that participates in the T3SS assemble, cannot replicate effectively into the plant tissues [14,15]. Instead, the T3SS channel allows the injection of bacterial proteins called type 3 secreted effectors (T3SEs). The T3SEs have diverse molecular functions that trigger the suppression of defense response and favor an aqueous apoplastic environment where pathogen fitness increases [16]. *Pst* DC3000 has described at least 36 functional T3SE with different molecular functions [17,18,19,20]. The T3SS and T3SE are genomically distributed in a tripartite pathogenicity island (T-PAI) that includes the *hrp-hrc* cluster (containing genes of the T3SS), the conserved effector locus (CEL) containing the conserved T3SE genes, and the Exchangeable Effector Loci (EEL) containing other virulence effectors that vary among strains [8]. The molecular function and targets of the T3SEs described so far are different and varied; however, they share the function of enhancing pathogen fitness during interaction with the host [19,20,21,22,23,24,25,26,27]. Additional to the T3SE, *P. syringae* produces phytotoxins that act as virulence factors [28]. Some examples of these molecules are coronatine, which mimics the plant hormone jasmonic acid, suppressing the defense hormone salicylic acid [29], phaseolotoxin that inhibits the arginine biosynthesis [30], among several others [31]. The accumulation, expression, and release of phytotoxins and T3SE facilitate the infective process of the pathogen.

Recently, the bacterial microbiota associated with Arabidopsis populations located in the southwest of France in ecologically contrasted habitats was explored and collected in fall and spring [32]. Together with the bacterial pathogens *Xanthomonas campestris* and *Pantoea agglomerans*, *P. syringae* was identified as one of the major components of the bacterial *A. thaliana* pathobiota (i.e., the ensemble of potentially pathogenic bacterial species). Among the *P. syringae* strains isolated, the *P. syringae* strain 0111-Psy-RAYR-BL (hereafter named *Psy* RAYR-BL) was described as highly virulent on the eight natural accessions tested [32]. Here, we report the genetic and pathogenic characterization of this strain. We found that *Psy* RAYR-BL belongs to the *P. syringae* Phylogroup 4 (PG4) [33] and can infect Arabidopsis and tobacco plants. The genomic analysis of *Psy* RAYR-BL revealed a reduced number of genes coding for T3SE than *Pst* DC3000. The HopBJ1 effector is coded in the *Psy* RAYR-BL genome but not in *Pst* DC3000. It is expressed in conditions that induce the bacterial virulence mechanisms. The plant expression of HopBJ1_Psy_ produces severe disease symptoms generating an increase in ROS accumulation and death of the tissue. Our study describes a novel mechanism of infection employed by PG4 *P. syringae* on natural Arabidopsis populations. This study is the first step for a better understanding of pathogenicity in strains with atypical T3E repertoires.

## 2. Materials and Methods

### 2.1. Plant Growth Conditions

*Arabidopsis thaliana* Col-0 ecotype, Jaco-C5 and Bazi-A2 [32], and *Nicotiana benthamiana* plants were grown during 4 weeks on peat: vermiculite substrate (1:1) with a 16 h light/8 h dark cycle at 22 °C (100 mmol m^−2^ s^−1^). 

### 2.2. Bacterial Culture and Plant Infections

*Pseudomonas syringae* pv *tomato* DC3000 or the *hrcC* mutated strain (*Pst* hrcC^−^) were grown on King B media supplemented with Rifampicin (50 μg/mL) and Kanamycin (50 mg/mL). *Pseudomonas syringae* RAYR-BL were cultured on King B media supplemented with Nitrofurantoin (20 ug/mL). *Agrobacterium tumefaciens* strain GV3101 was cultured on LB media supplemented with Rifampicin (50 μg/mL), Spectinomicin (50 μg/mL), and Gentamicin (30 μg/mL). All strains were grown at 28 °C for 24 h before plant inoculation. Then the strains were sedimented by centrifugation and resuspended in 10 mM MgCl_2_. The bacteria were inoculated by syringe infiltration on the abaxial side of the leaves using a suspension of 1 × 10^7^ colony forming units (CFU)/mL (for Arabidopsis assays) or 1 × 10^6^ CFU/mL (for tobacco assays). For bacterial proliferation assays, leaf discs (5 mm^2^) from 10 independent inoculated plants were taken at 0, 24, and/or 48 h post-inoculation. The tissue was ground in 10 mM MgCl_2_, serially diluted 1:10, and plated onto King B agar plates supplemented with the corresponding antibiotics. Plates were incubated at 28 °C for 2 days, and the Colony Forming Units (CFU) were estimated. Accumulation of H_2_O_2_ was detected using 3-3’ diaminobenzidine (DAB) following the protocol described in [34]. Each bacterial proliferation and related assays were repeated three times.

### 2.3. Comparative Genomics, Gene Ontology Analysis, and Functional Annotation

Comparison between the Reference *Pst* DC3000 [28] and *Psy* RAYR-BL [35] genomes was performed using BLAST Ring Image Generator (BRIG) v0.95 (available at https://sourceforge.net/projects/brig/, accessed on 30 November 2021) [36], with an e-value of 1 x 10^−6^ and colored according to BLAST identity of 100%, 70% and a minimum threshold of 50%. Protein coding sequences in genomic regions of *Pst* DC3000 with low similarity to *Psy* RAYR-BL and lengths above 10 kb were further analyzed. Identification of orthologous genes across the genomes was made with OrthoFinder v2.3.1 (available at https://github.com/davidemms/OrthoFinder, accessed on 30 November 2021) [37] with default parameters. The GO enrichment, and the cluster of orthologous groups (COG) were determined by EggNOG mapper v2.0.1 (available at http://eggnog-mapper.embl.de/, accessed on 30 November 2021) [38], and the KEGG pathways analysis was performed on the KEGG Automatic Annotation Server (KAAS, available at https://www.genome.jp/kegg/kaas/, accessed on 30 November 2021) [39]. Metabolic pathways were obtained using the KEGG Mapper-Reconstruct Pathway v.3.2 (available at https://www.genome.jp/kegg/mapper.html, accessed on 30 November 2021). The circular genome representation of *Psy* RAYR-BL was generated using CGView v2 (available at https://toolkit.tuebingen.mpg.de/gcview, accessed on 30 November 2021) [40] based on the visualization of CDS and sequence feature information.

### 2.4. Phylogenomics Analysis

A maximum likelihood (ML) phylogenomic analysis was performed based on the concatenated protein sequences of 1428 single-copy orthologs (397,426 positions) determined by OrthoFinder [37] on the genomes of 25 strains of *Pseudomonas syringae,* including *Psy RAYR-BL* (representing 11 phylogroups) and *Pseudomonas aeruginosa* PAO1 used as an outgroup. The sequences were aligned using Multiple Alignment using Fast Fourier Transform (MAFFT) v7.271 [41] with maximum iterative refinement set to 1000, and Gblocks v0.91b [42] was used to trim uninformative positions. The ML phylogenomic tree was built using Randomized Axelerated Maximum Likelihood (RaxML v8.2.12) [43] with 1000 rapid bootstrap replicates with the JTT+I+G (JonesTaylor Thornton + invariant sites + gamma distribution) amino acid substitution model determined by ProtTest v3.4.2 [44] under the AIC criterion. A similar procedure was followed for the Multilocus sequence typing (MLST) phylogeny based on nine housekeeping genes using the GTR+I+G (General Time Reversible + invariant sites + gamma distribution) nucleotide substitution model determined by jModelTest v2.1.10 [45]. Accession numbers are reported in Appendix A. 

### 2.5. Identification of Type III Secretion System and Other Virulence-Associated Genes

The presence of type III effector proteins was determined using a similar protocol to the one described by Martínez-García et al. [20], with a BLASTP search of the effector proteins in http://pseudomonas-syringae.org/ (accessed on 30th November 2021) with an e-value of 1 × 10^−6^, selecting the one with the best identity and coverage when a protein sequence matched several T3SEs. We also considered an effector protein as incomplete if the alignment length was ≥25% smaller than the length of the reference T3SE. Also, T3SE were predicted using the EffectiveDB (https://effectors.csb.univie.ac.at/, accessed on 3 January 2022) [46] and the BEAN 2.0 (http://systbio.cau.edu.cn/bean/, accessed on 3 January 2022) [47] web tools. To identify homologous gene clusters coding for the type III secretion system (T3SS), and for the phytotoxins coronatine, phaseolotoxin, mangotoxin, tabtoxin, syringolin A, syringomycin, and syringopeptin, MultiGeneBlast v1.1.14 with default parameters was used. The cluster was considered present if at least half the genes were found. As a reference for the T3SS and coronatine gene cluster, *P. syringae* DC3000 [28] was used. *Pseudomonas syringae* pv. *oryzae* 1_6. *P. syringae* pv. *phaseolicola* 1448A, *P. syringae* pv. *tabaci* BR2, and *P. syringae* pv. *syringae* UMAF0158 were used as references for phaseolotoxin, tabtoxin, and mangotoxin, respectively. *P. syringae* pv. *syringae* B728a was used for syringomycin and syringopeptin. Other virulence factors were screened using the VFAnalyzer pipelinee of the Virulence Factor DataBase (VFDB) (accessed on 20 November 2021) [48].

### 2.6. HopBJ1_Psy_ Mutation and Nicotiana Transient Expression by Agroinfiltration

The *hopBJ1_Psy_* gene was amplified by PCR using genomic *Psy* RAYR-BL DNA as a template and the oligonucleotides listed in Appendix A. The amplicon was cloned on the vector pENTR-D-TOPO and then LR-recombined with the plasmid pGWB405 (Addgene #74799, Wattertown, MA, USA) according to manufacturer indications (Gateway™ LR Clonase™ II Enzyme mix, ThermoFisher, Whaltham, MA, USA). The site-directed mutation was generated by overlap extension PCR using oligos listed in Appendix A. The mutated amplicon was cloned into the vector pENTR-D-TOPO. All constructs were sequenced at Macrogen, INC (Seoul, Korea). For Agrobacterium infiltration of tobacco plants, chemically competent Agrobacterium bacteria were transformed with vectors that contain the generated 35S:HopBJ1_Psy_-GFP construct, the mutated construct, and the non-related 35S:WRKY7-GFP construct as control. The transient expression was performed as described on [49].

## 3. Results

### 3.1. The Strain Pseudomonas syringae RAYR-BL Induces Severe Disease on A. thaliana 

The *Psy* RAYR-BL genome amplifies for a molecular marker used to rapidly identify strains in the *Pseudomonas syringae* group [50] (Appendix A). When grown on King’s B media, the strain produces fluorescent molecules, a characteristic of several *Pseudomonas syringae* strains [35,51], including *P. syringae* pv *tomato* (*Pst*) DC3000 used in our study as a reference Pseudomonas pathogenic model. These results support this recently discovered strain classification as belonging to the *Pseudomonas syringae* species. To characterize the growth kinetics of *Psy* RAYR-BL, we evaluated bacterial growth on King’s B media. We did not detect differences in the kinetic parameters of *Psy* RAYR-BL compared to *Pst* DC3000 (Appendix A).

*Pst* DC3000 induces the Effector Triggered Susceptibility (ETS) response on plants, displaying a disease phenotype characterized by chlorosis, uncontrolled bacterial proliferation in the inoculated tissue, and the eventual death of tissue [52]. On the other hand, the isogenic *P. syringae* hrcC^−^ strain lacks the *hrcC* gene required to release virulence effectors [14,15]. The *Pst* hrcC^−^ strain triggers the PAMP-Triggered Immunity (PTI) by the plant recognition of bacterial PAMPs and the inability of the strain to produce an active proliferation.

The inoculation of the Arabidopsis accessions BAZI-A-2 and JACO-C-5 found in natural habitats in the southwest of France [32] with the *Psy* RAYR-BL strain induces chlorosis as a clear disease symptom (Figure 1A). Unlike *Pst* hrcC hrcC^−^, the *Psy* RAYR-BL strain grows in the plant tissue, and the bacterial titer increases 48 h post-inoculation (hpi), like the virulent strain *Pst* DC3000. Even when both strains proliferate in the plant tissue, *Pst* DC3000 has an increased titer compared to the *Psy* RAYR-BL (*p* < 0.01 on all the evaluated ecotypes according to an ANOVA/Tukey’s test) (Figure 1B). This result suggests a plant susceptibility response. To evaluate ROS accumulation, the inoculated Arabidopsis leaves were stained with 3,3’-Diaminobenzidine (DAB) that is oxidized by H_2_O_2_, thereby generating a dark brown precipitate [34]. The sites inoculated with the *Psy* RAYR-BL show the DAB characteristic staining mark at 24 hpi, like the virulent *Pst* DC3000 strain. In contrast, the *Pst* hrcC^−^strain does not produce a detectable accumulation of precipitated compared to the mock-inoculated leaves (Figure 1C). We then tested the infective process of *Psy* RAYR-BL on the model *A. thaliana* accession Columbia-0 (Col-0). The inoculation of Col-0 with *Psy* RAYR-BL displays similar phenotypes compared with the natural Arabidopsis accessions showing no visible phenotypical differences or differences in the ROS accumulation (Appendix A). The quantitation of bacterial proliferation does not show statistical differences (*P* = 0.4842 in a two-way ANOVA analysis) among the accessions (Figure 1B).

These results indicate that the *Psy* RAYR-BL strain induces a susceptibility response in *Arabidopsis thaliana*, including natural accessions like BAZI-A-2 and JACO-C-5 and the model Col-0 accession.

### 3.2. Comparative Genomics of Pseudomonas syringae DC3000 and Pseudomonas syringae RAYR-BL

Due to the similar pathogenicity behavior displayed on Arabidopsis plants [32], we investigated the genomic features between the *Psy* RAYR-BL strain (isolated from natural *A. thaliana* populations) and *Pst* DC3000 (isolated from tomato [53]). The *Psy* RAYR-BL draft genome [35] has a size of 5.85 Mbp assembled in 110 contigs (N50: 193.008), slightly smaller than the *Pst* DC3000 genome that reaches 6.54 Mbp. The GC% is 58.89% for *Psy* RAYR-BL, very close to the 58.4% described for the *Pst* DC3000 used as the reference genome for assembly and comparative analysis [28]. *Psy* RAYR-BL includes 5188 genes, of which 5053 have an open reading frame, which ranges between the gene count previously found for the *P. syringae* group (between 4973–6026 genes, average 5465) [54]. The general features of the sequenced genome and the reference genome [28] are listed in Table 1. The circular genome map of *Psy* RAYR-BL, location of genes in both directions, Clusters of Orthologous Groups of proteins (COG), and virulence factors are displayed in Figure 2. Detailed genome annotation and virulence factors analysis data can be found in Appendix A.

As a first approach to detect differences between *Pst* DC3000 and *Psy* RAYR-BL, the genomes were aligned with BRIG [36]. As expected, the BRIG genome circle generated with *Pst* DC3000 used as a reference shows low similarity or absent regions (gaps) on the *Psy* RAYR-BL. We detected 31 genomic regions with a length greater than 10 Kb, absent in the *Psy* RAYR-BL genome (Figure 3A). These regions comprise a total of 388 CDSs, including 111 hypothetical proteins, mobile genetic elements, kinases, and TS3E, such as hopAD1 (Locus_tag: PSPTO_RS24270), hopQ1-1 (Locus_tag: PSPTO_RS04650), and AvrPto1 (Locus_tag: PSPTO_RS20745) (Appendix A). Putative orthologous genes were identified with OrthoFinder [37] (Figure 3B and Appendix A) to determine the difference in gene functions on the genomes. The analysis classified 9303 genes into 4054 orthogroups. In total, 4538 (89.8% of the strain CDSs) and 4765 (80.9% of the strain CDSs) genes correspond to *Psy* RAYR-BL and *Pst* DC3000, respectively, and 3925 out of the 4054 identified orthogroups are shared between the strains. Of these, 3365 are single-copy orthologues. As well, 129 orthogroups are strain-specific, 28 correspond to *Psy* RAYR-BL (78 genes), and 101 to *Pst* DC3000 (241 genes). Furthermore, 1167 are classified as unassigned genes (genes that cannot be assigned to any orthogroup), 515 belong to *Psy* RAYR-BL, and 652 to *Pst* DC3000. In total, 593 genes (78 genes in 28 strain-specific orthogroups plus 515 unassigned genes) are exclusively present on the *Psy* RAYR-BL genome. In comparison, 893 genes are absent on this strain but present on *Pst* DC3000 (241 genes in 101 strain-specific orthogroups plus 652 unassigned genes) (Figure 3B, Appendix A). Among this group of genes that is not shared, we identified some virulence-associated proteins, such as IucA/IucC, involved in siderophore biosynthesis (Locus_tag: K0038_02720 and Locus_tag: K0038_02723) [55], and virulence-associated protein E, vapE [56]. Nevertheless, these groups of genes likely correspond to hypothetical proteins, thereby potentially explaining that despite these differences in gene number, no considerable differences were revealed in functional COG, KEGG, and GO analyses (Figure 3C–E and Appendix A).

### 3.3. Phylogenetic Position of Pseudomonas syringae RAYR-BL 

A phylogenomic tree was constructed based on the core proteome of *Psy* RAYR-BL and 25 strains representative of 11 of the 13 phylogroups of the *Pseudomonas syringae* complex, as well as *Pseudomonas aeruginosa* PAO1 used as outgroup, comprising a total of 1428 sequences and 397,426 positions. Previous analyses, also considering the core proteome of varying Pseudomonas strains, identified different numbers of core sequences. For example, Nikolaidis et al. [54] defined the *P. syringae* group based on 2944 proteins and 110,643 positions, considering 34 assemblies only annotated as "complete". Other studies defined the *P. syringae* group based on 2743 [57], 3397 [58], and 343 [59] core sequences estimated from 13, 19, and 127 genomes, respectively. Thus, our analysis corresponds well with previous studies considering the differences arising from our selection of representatives of 11 of the 13 phylogroups of the *P. syringae* complex and those resulting from variations in methodology, the assembly level of the genomes (complete, scaffold, contig), full or single-copy orthologue gene sets, and the outgroup genome selection. The resulting maximum likelihood phylogenomic tree also agrees with the delimitation of the phylogroups previously described [33,60] and grouped *Psy* RAYR-BL in the canonical *Pseudomonas syringae* lineage belonging to the Phylogroup (PG) 4 (Figure 4). Multilocus sequence typing (MLST) phylogeny based on nine housekeeping genes gave congruent results (Appendix A). This classification is based on some genetic characteristics of the late branches of Pseudomonas lineages such as the inoculum isolation from plants and the presence of a tripartite pathogenesis island (T-PAI).

### 3.4. Analysis of Molecular Components That Participate in the Pseudomonas syringae Pathogenesis

We evaluated the genetic components that can confer the pathogenic potential of the *Psy* RAYR-BL strain. In particular, we focused on genes that could play a role in the host-pathogen interaction. We considered the evaluation of genes that participated in the structure and function of the type III secretion system (T3SS) necessary for the translocation of virulence effectors from the bacteria to the host cell. The comparison between the *hrp–hrc* gene cluster that encodes structural components and regulators of the T3SS shows that, as for *Pst* DC3000, *Psy* RAYR-BL strains possess the conserver effector locus (CEL) and the exchangeable effector locus (EEL) (Figure 5A, Appendix A). In addition, using a sequence analysis approach based on alignment, domain signals, and machine-learning predictions (EffectiveDB and BEAN2.0 bioinformatics services) [46], we evaluated the non-conserved genes present in *Psy* RAYR-BL and absent in *Pst* DC3000 and found in both the CEL and EEL regions putative new effectors never described in the *P. syringae* complex [46,47] (Appendix A). We also evaluated the genetic potential to produce phytotoxins. These secondary metabolites participate in the colonization process, facilitating the disease progression [28,29,30,31]. We considered a potential capacity of metabolite production if the genome includes at least 50 percent of the genes described as required for the compound biosynthesis. Accordingly, we found that the *Psy* RAYR-BL strain possesses the capacity to produce phaseolotoxin, mangotoxin, syringomycin, and syringopeptin (Table 2, Appendix A). Furthermore, we experimentally demonstrate that the *Psy* RAYR-BL strain can produce syringomycin/syringopeptin, unlike *Pst* DC3000 (Appendix A).

Additionally, to evaluate the known effectors repertoire in *Psy* RAYR-BL, we evaluated the gene presence of type 3 secreted effector proteins (T3S) based on sequence identity with the previously described effector found on the Pseudomonas-Plant Interaction Resource database (www.pseudomonas-syringae.org, accessed on 30 November 2021). The analysis detects 40 T3SE genes on the *Pst* DC3000 genome, including two partial gene sequences. On the other hand, we detect only six T3SE genes in the *Psy* RAYR-BL genome, two with a partial sequence and four with a complete sequence. These four T3SE complete genes encode for *avrE1_Psy_*, *hopB1_Psy_*, *hopAH1_Psy_,* and finally *hopBJ1_Psy_,* which is only present on the *Psy* RAYR-BL genome and not in the *Pst* DC3000 genome (Figure 5B, Appendix A).

### 3.5. hopBJ1_Psy_ Is a Virulence Factor Required for Psy RAYR-BL to Produce Plant Disease

Surprisingly, despite the pathogenic phenotype displayed by the *Psy* RAYR-BL strain, this strain harbors a reduced repertoire of virulence effectors compared to the model *Pst* DC3000 strain (Figure 5B). Among the four effectors classified as complete sequences coded in the *Psy* genome, the gene *hopBJ1_Psy_* stands out because of its absence on the *Pst* model genome. The *hopBJ1_Psy_* gene codes for a protein with 254 amino acids that share the 84.25% identity with the HopBJ1 protein identified on the *P. syringae* strain CC1557 belonging to the phylogroup 10 of the *P. syringae* complex. The protein possesses the conserved amino acids Cys153 and His171 required for its function [61] (Figure 5C). We determined its expression on a nutritional deficit to evaluate whether the *hopBJ1_Psy_* gene is functional (Figure 6). The expression of virulence effectors is induced in poor bacterial media [62,63]. Accordingly, the expression of *hopBJ1_Psy_* is induced 2.4-fold when *Psy* RAYR-BL is incubated in a saline solution without organic carbon sources compared to a rich media (LB, Figure 6A). Additionally, we measured the expression of the *hopBJ1_Psy_* transcript on plant tissue infected with the *Psy* RAYR-BL strain (Figure 6B). In these conditions, we detected the presence of the bacterial mRNA, indicating that this gene is expressed during the interaction between *Psy* RAYR-BL and plants in vivo.

The description of *P. syringae* CC1557 and the participation of HopBJ1 on pathogen virulence on tobacco plants were previously reported [61]. Therefore, we evaluated the *Psy* RAYR-BL disease phenotype on this species as a plant host.

*N. benthamiana* plants were inoculated by infiltration with *Pst* DC3000, *Pst* hrcC^−^ or *Psy* RAYR-BL strains. *Psy* RAYR-BL produces slightly visible symptoms after 24 hpi that are clearly detectable after 48 hpi (Figure 7). The inoculation with *Pst* DC3000 produces visible symptoms at 48 hpi that increase after 72 hpi. On the other hand, inoculation while *Pst* hrcC^−^ does not display a disease phenotype (Figure 7A). Correlated with the described phenotypes, the *Psy* RAYR-BL strain has a high proliferation on Nicotiana plants, reaching 2.2 orders of magnitude of growth after 72 hpi. This increase is statistically significant compared to the other evaluated strains (*p* < 0.0001, Figure 7B). We also detected a strong DAB stain at the sites inoculated with the *Psy* RAYR-BL at eight hpi. At the same time, the other strains did not produce a detectable accumulation of precipitates compared to the non-inoculated leaves. *Pst* DC3000-inoculated leaves display a slight stain accumulation compared with the *Psy* RAYR-BL. The inoculation with *Pst* hrcC^−^ does not produce a detectable oxidized DAB (Figure 7C). Considering these results, we conclude that the *Psy* RAYR-BL strain is a highly aggressive *Pseudomonas syringae* strain that produces disease on Arabidopsis plants and *Nicotiana benthamiana*.

As a simple approach to evaluate the effect of HopBJ1*_Psy_* on the disease phenotype displayed by *Psy* RAYR-BL, the gene was cloned and fused to the GFP gene (HopBJ1_Psy_-GFP) and transitorily expressed under the control of the CaMV35S promoter on *N. benthamiana* leaves by Agroinfiltration (Figure 8). As a negative control, we use untransformed Agrobacterium and a vector carrying the construct that includes a non-related protein fused to GFP (WRKY7-GFP). After 48 h of Agroinfiltration, the HopBJ1_Psy_-GFP-inoculated leaves showed a severe phenotype characterized by necrotic HR-like lesions. The inoculated leaves with the empty and the WRKY11-GFP-carrying Agrobacterium do not display a visible phenotype (Figure 8A). Despite the lesion of leaves expressing the construct HopBJ1*_Psy_*, Agrobacterium does not change the proliferation rate compared to the plants that do not express the transgene (Appendix A).

The HopBJ1 protein shares a similar structure with the CNF1 toxin of *E. coli*. CNF1 possesses two conserved amino acids required for catalytic function [61,64]. These amino acids are also present in the HopBJ1*_Psy_* protein (Cis153 and His171). To test the participation of these amino acids on the HopBJ1*_Psy_* function, we generated a mutated version of the HopBJ1 (HopBJ1^C153S/H171A^) that was further heterologously expressed in tobacco plants (Appendix A). We observed that the expression of the mHopBJ1 does not generate a visible phenotype on tobacco plants (Figure 8B), indicating that the amino acids C153 and H171 are required for the protein function.

## 4. Discussion

A previous study described pathogens from natural Arabidopsis populations from North America [65]. The classification of the strains in this study shows two major pathogenic species found in Arabidopsis: the *P. viridiflava* Phylogroup (PG) 7 and 8 and the *P. syringae* strains belonging to PG2 of the *P. syringae* species complex [33,65,66]. Similarly, on *A. thaliana* populations located in the southwest of Germany, the most abundant Pseudomonas operational taxonomic unit was the OTU5 affiliated to *P. viridiflava* [67]. According to our phylogenetic analysis, we considered the core proteome of 25 representative strains from the *P. syringae* complex, the *Psy* RAYR-BL strain clusters in the PG4 (Figure 4). Few strains have been described previously in this PG. They have been isolated from different habitats such as the plants *Hutchinsia alpina*, oats [66], rice [58], and also from snow [33]. Thus, no previous report described PG4 strains from Arabidopsis nor capable of producing disease on this plant species. Our study establishes the PG4 as an emerging pathogenic lineage for *A. thaliana* by claiming concerns on the emergence of novel Brassicaceae pathogens.

All the previous strains described on the PG4 produce an HR phenotype on tobacco, a characteristic shared with the strain *Psy* RAYR-BL. Nevertheless, unlike strains belonging to the PG4, according to the blastp search of known T3SE, the genome of *Psy* RAYR-BL does not code for the T3E genes *hopBH1* and *hopBI1* [68].

Despite its strong pathogenicity on Arabidopsis and tobacco, the *Psy* RAYR-BL possess a small repertoire of T3E. This genomic feature is shared with strains that cluster in PG7 that have also been associated with a disease phenotype on Arabidopsis plants [66,69]. The high virulence and small repertoire of T3SE could be explained by the presence of the AvrE1 gene on the *Psy* RAYR-BL genome because the presence of this virulence effector in PG7 correlates with the pathogenicity of the strains with a reduced number of T3SE [70]. Additionally, the *Psy* RAYR-BL genome codes for several genes belonging to the biosynthetic pathways of phytotoxins that favor bacterial virulence (Table 2). The *Psy* RAYR-BL can potentially produce phaseolotoxin and mangotoxin, and we experimentally demonstrate that it can produce syringomycin/syringopeptin (Table 2, Appendix A). The reported description agrees with the described correlation between the presence of the syringomycin gene cluster and the small T3E repertoire [58].

The *Psy* RAYR-BL codes for four complete T3E genes: *avrE1_Psy_*, *hopB1_Psy_*, *hopAH1_Psy_*, and *hopBJ1_Psy_*. The *avrE1_Psy_*, *hopB1_Psy_*, and *hopAH1_Psy_* genes are also coded on the *Pst* DC3000 genome, while *hopBJ1_Psy_* is only present in the *Psy* RAYR-BL genome (Figure 5B). We showed the induction of the *hopBJ1_Psy_* expression when *Psy* RAYR-BL is exposed to low nutrient conditions. Additionally, we demonstrated that the effector is expressed in the interaction of the strain with the host plant (Figure 6). Accordingly, the low availability of nutrients and the bacterial interaction with the host have been described as conditions that induce virulence in *P. syringae* [62,63]. These results support the hypothesis that the *hopBJ1_Psy_* is a functional gene and could act as a virulence factor.

The plant expression of the *hopBJ1_Psy_* gene in plants has a detrimental effect on plant cell survival, inducing a disease phenotype by increasing ROS accumulation and inducing cell death in tobacco (Figure 8A). Previously, Hocket et al. [61] described the HopBJ1 protein from the *P. syringae* strain CC1557 strain as a T3ES that has sequence similitude with the CNF1 toxin from *E. coli* [61,64]. We demonstrated that the mutation on the HopBJ1*_Psy_* amino acids conserved on the CNF1 active site abolished the cell death observed when the tobacco plants express the wild-type HopBJ1*_Psy_* (Figure 8B). The *E. coli* CNF1 protein interferes with glutamine deamination affecting the actin cytoskeleton in animal cells [64]. Due to the similarities of these proteins, we hypothesize that they could act with a similar biochemical mechanism. Several studies pointed to the actin cytoskeleton as an active factor in cell death and programmed cell death (PCD) on plants [71,72,73]. Besides, the *P. syringae* virulence effectors HopG1 and HopW1 target the actin cytoskeleton [74,75,76]. Thus, HopBJ1*_Psy_* could produce the observed phenotype by affecting the cytoskeleton dynamics. However, further analysis is required to determine its molecular function and targets.

Interestingly, in addition to the genes coding for virulence factors found on the *Psy* RAYR-BL genome, by using the BEAN 2.0 [47] and the EffectiveDB web tools [46], we detect sequences contained on the CEL and EEL that have, for example, domains only found on T3SEs or type III secretion signals, thus predicted to be a putative T3SE (Appendix A). Furthermore, both tools detected HopBJ1 and detected genes identified as type III helper proteins. Therefore, further sequence and functional analysis of these genes would help confirm or discard these predictions and provide new information on the virulence mechanisms underlying the Pseudomonas pathogenesis.

Our study is a first step in understanding novel mechanisms associated with the pathogenicity of emerging *P. syringae* strains from wild *A. thaliana* populations. Future investigations on the putative effectors found in strains with reduced T3SS will help to understand the molecular processes underlying disease emergence, and by consequence, anticipate disease control strategies. Our study also corroborates the importance of monitoring non-agricultural habitats as reservoirs of newly emerging pathogens.

## Figures and Tables

**Figure 1 microorganisms-10-00707-f001:**
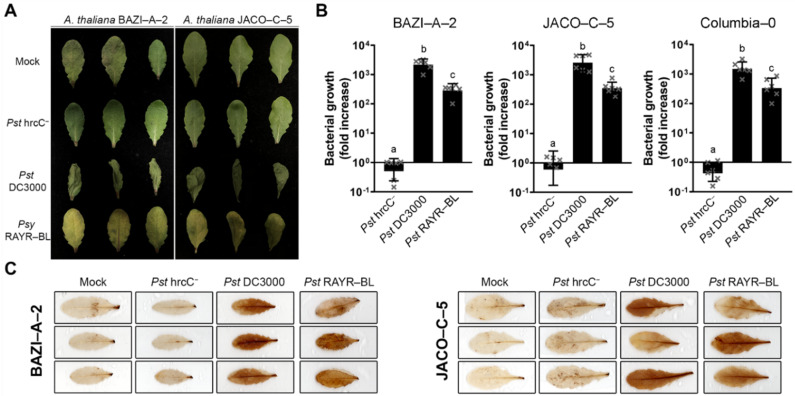
Arabidopsis accessions obtained from the field are susceptible to infection with the *Pseudomonas syringae* RAYR-BL strain. Four-week-old BAZI-A2, JACO-C5 and Columbia-0 plants were infiltrated with *Pseudomonas syringae* RAYR-BL (*Psy* RAYR-BL), *Pseudomonas syringae* pv *tomato* DC3000 (*Pst* DC3000), *Pseudomonas syringae* hrrC− (*Pst* hrcC^−^), or MgCl_2_ 10 mM as control (Mock). (**A**) The disease phenotype of BAZI-A2 and JACO-C5 plants was visualized 48 h post-inoculation (hpi). (**B**) Bacterial proliferation was evaluated on BAZI-A2, JACO-C5, and Columbia-0 plants by counting the colony-forming units on plant tissue and the mean of bacterial growth since the inoculation ± SD (*n* = 6 represented as little x in the bars). The statistical comparison between the different strains displays a significant difference for each comparison according to a 2-way ANOVA analysis and Tukey’s post-test *p* < 0.01The different letters above bars indicate these significant differences. There was no statistical difference between the ecotypes according to a two-way ANOVA (*p* = 4842). (**C**) H_2_O_2_ levels on BAZI-A2 and JACO-C5 plants were detected using DAB staining on the inoculated leaves after 48 hpi.

**Figure 2 microorganisms-10-00707-f002:**
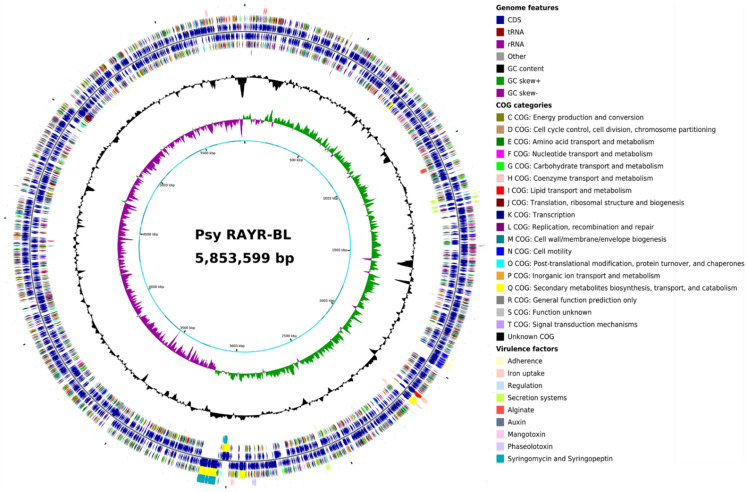
Circular genome map of *Psy* RAYR-BL created with CGView. Moving from the outermost ring: (**1**) potential virulence factors screened with the VFDB VFAnalyzer tool and searched with MultigeneBlast on forward strand, (**2**) CDS on forward strand colored according to COG category, (**3**) CDS, tRNA, and rRNA on forward strand, (**4**) CDS, tRNA, and rRNA on reverse strand (**5**), CDS on reverse strand colored according to COG category, (**6**) potential virulence factors screened with the VFDB VFAnalyzer tool and searched with MultigeneBlast on reverse strand, (**7**) GC content, (**8**) GC skew, and (**9**) draft genome position in Mbp with contig boundaries colored in light and dark cyan.

**Figure 3 microorganisms-10-00707-f003:**
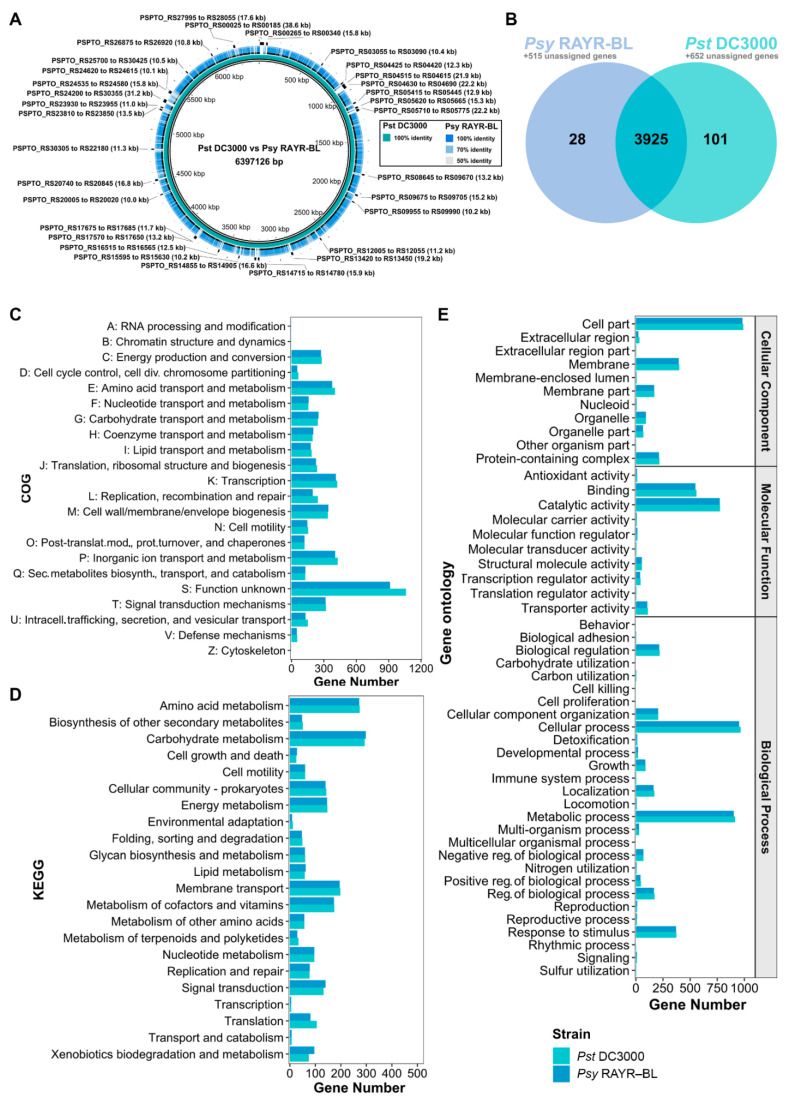
Whole-genome and proteome comparison between *Pst* DC3000 and *Psy* RAYR-BL. (**A**) Blastn comparison of the draft genome of *Psy* RAYR-BL against *Pst* DC3000 with the BRIG software. White spaces indicate regions of the genomes with similarity <70%. Genes in regions of length ≥10 kb absent in *Psy* RAYR-BL are highlighted in black. (**B**) Venn diagram of the orthologous genes predicted by gene clustering using OrthoFinder. (**C**–**E**) Comparison of the proteomes of *Psy* RAYR-BL and *Pst* DC3000 according to their COG category (**C**), KEGG pathway (**D**)**,** and GO term (**E**).

**Figure 4 microorganisms-10-00707-f004:**
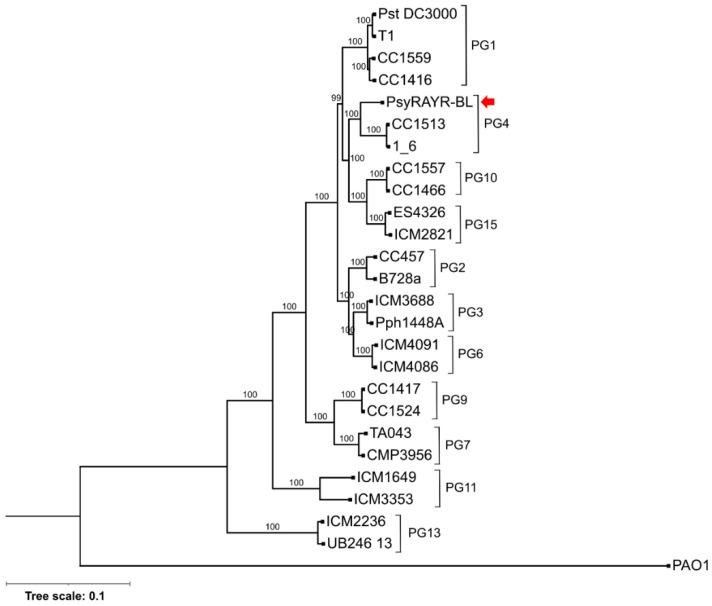
Phylogenetic relationships of *Psy* RAYR-BL with other *P. syringae* strains. Phylogenomic Maximum Likelihood tree was estimated based on the partial amino acidic sequence of 1428 concatenated single-copy orthologues genes determined by OrthoFinder with a total of 397,426 positions in the final dataset. Maximum-likelihood phylogenomic analysis was performed based on the concatenated protein sequences of single-copy orthologues determined by OrthoFinder on genomes of 26 strains of *Pseudomonas syringae* (representing 11 phylogroups). *Pseudomonas aeruginosa* PAO1 was used as an outgroup. The red arrow points the *Psy* RAYR–BL strain.

**Figure 5 microorganisms-10-00707-f005:**
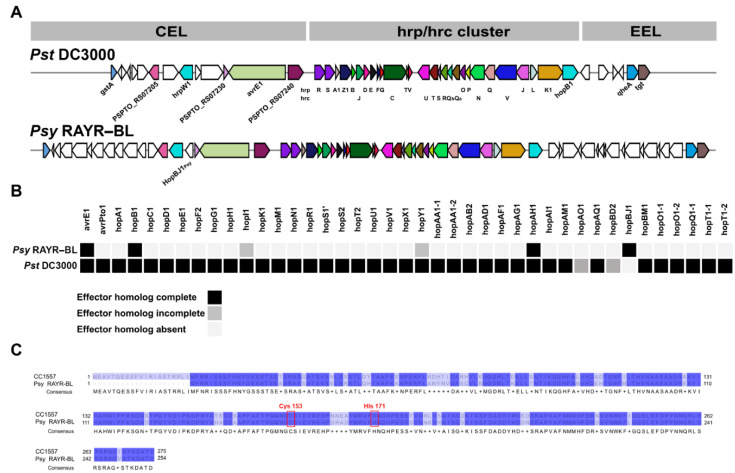
Organization of the type III secretion system (T3SS) and repertoire of effector proteins. (**A**) Analysis of the T-PAI (tripartite pathogenicity island) organization of the T3SS, which includes the *hrp/hrc* gene cluster and the flanking CEL (conserved effector locus) and EEL loci (exchangeable effector locus). Comparison among *Pst* DC3000 and *Psy* RAYR-BL was made using MultiGeneBlast. (**B**) Type 3 effector proteins (T3S) across *Pst* DC3000, *Psy* RAYR-BL. Black boxes indicate the presence of the full-length protein, gray boxes indicate an incomplete alignment (alignment length <25% of T3SS reported length), and light gray boxes indicate no significant matches (e-value > 1 × 10^−6^). (**C**) Amino acidic alignment between the HopBJ1 from *P*. *syringae* CC1557 and HopBJ1*_Psy_* from *Psy* RAYR-BL. Red boxes indicate amino acids described as essential for protein function.

**Figure 6 microorganisms-10-00707-f006:**
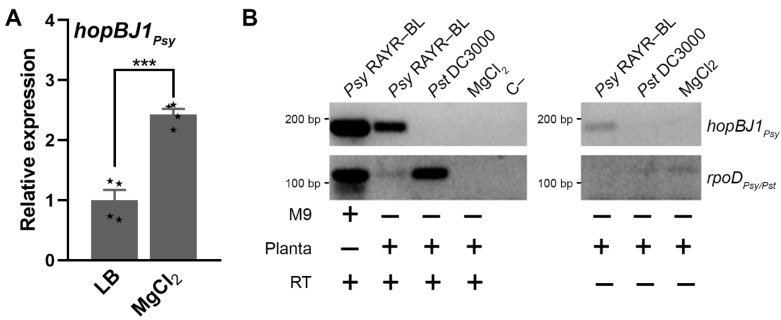
The *hopBJ1**_Psy_* gene is expressed in conditions where bacterial virulence is induced. (A) The bacterial expression of *hopBJ1**_Psy_* was evaluated in a medium without nutrients. *Psy* RAYR-BL strain grown on King’s B media, and then it was re-suspended on MgCl_2_, or King’s B media as control. Samples were incubated for one hour, and then samples were collected and frozen. The data are expressed as folds of induction of the *hopBJ1**_Psy_* gene relative to the King’s B media condition ± SE (*n* = 3 indicated as small asterisks in the bars). The *rpoD* gene was used as the housekeeping gene. The asterisks between the bars (***) indicate significant differences between the conditions according to a *t*-test (*p* < 0.001). (B) The expression of *hopBJ1_Psy_* was evaluated during the plant-pathogen interaction. Plants were inoculated with *Psy* RAYR-BL, *Pst* DC3000, or MgCl_2_ as control. After 3 h, the inoculated leaves were sampled and frozen on liquid N_2_. RNA was extracted and treated with DNAse I to discard the possible contamination with genomic DNA on the sample. Then, PCR reactions were performed to detect the genes *hopBJ1* and *rpoD* as a bacterial housekeeping gene to detect the presence of bacterial RNA on the sample. As a negative control, RNA (RT −) or water (C −) was used as the template for the PCR reaction.

**Figure 7 microorganisms-10-00707-f007:**
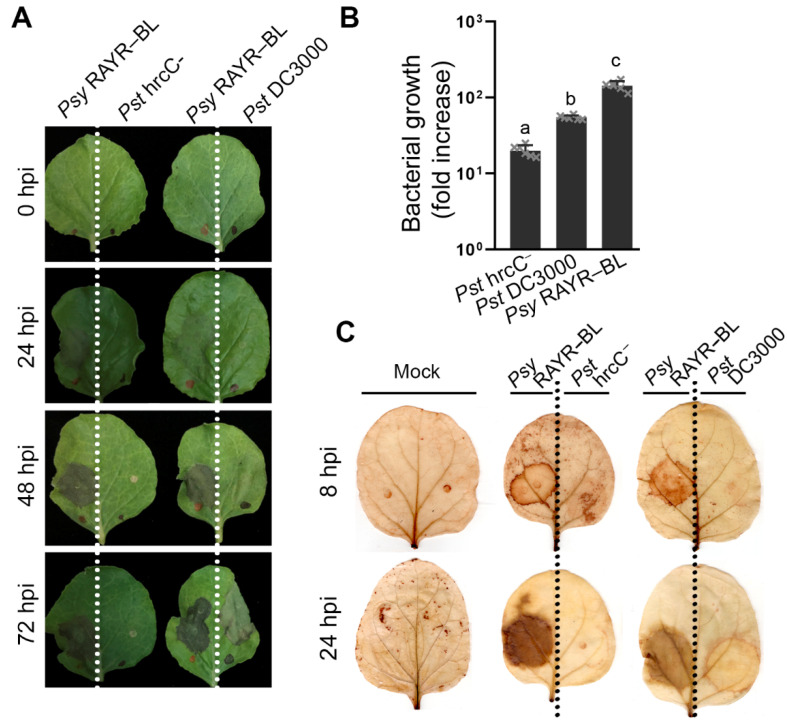
*Pseudomonas syringae* RAYR-BL strain produces a severe disease phenotype on tobacco plants. Four-week-old plants were syringe-inoculated with *Psy* RAYR-BL on the left side of the leave. The right side was inoculated with *Pst* hrcC^−^ or *Pst* DC3000. (A) The disease phenotype was visualized 72 h post infiltration (hpi). (B) In planta*,* bacterial proliferation was evaluated by counting the colony-forming units on plant tissue after 48 hpi. The data are expressed as the mean of bacterial growth since the inoculation ± SD (*n* = 6 represented as little gray x in the bars). The statistical comparison between the different strains displays a significant difference for each comparison according to a 2-way ANOVA analysis and Tukey’s post-test *p* < 0.01The different letters above bars indicate these significant differences). The different letters above bars indicate significant differences between the conditions according to a two-way ANOVA analysis and Dunnett’s post-test (*p* < 0.0001). (**C**) H_2_O_2_ levels were detected using DAB staining on the inoculated leaves. Control leaves were infiltrated with MgCl_2_ 10 mM (Mock).

**Figure 8 microorganisms-10-00707-f008:**
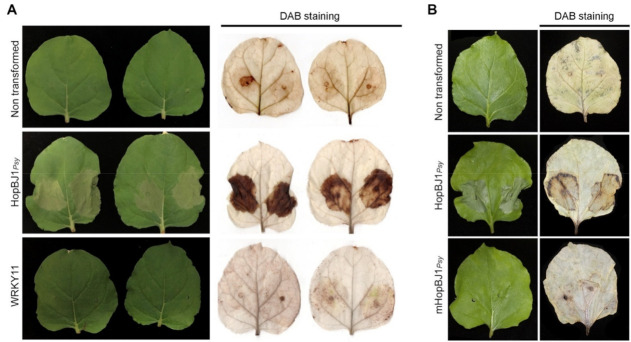
The expression of HopBJ1***_Psy_*** induces the death of the plant tissue and the amino acids Cys153 and His171 are required to generate the plant phenotype. Tobacco plants were agro-infiltrated with Agrobacterium carrying the *hopBJ1**_Psy_* controlled by a CaM35S promoter in a wild-type version (HopBJ1*_Psy_*). As a control, untransformed Agrobacterium or an Agrobacterium strain carrying a non-related construct (WRKY11-GFP) were inoculated. Two days post-inoculation, plants were photographed (A, left panel) and stained with DAB for ROS detection (A, right panel). To evaluate the participation of the amino acids Cys174 and His192 a mutated version of HopBJ1 (Cys153Ser and His171Ala, mHopBJ1*_Psy_*) was expressed on tobacco plants (B). Plants were photographed two days post-Agro infiltration (left panel). Inoculated leaves were stained with DAB for ROS detection (right panel).

**Table 1 microorganisms-10-00707-t001:** Summary of the assembly and annotation characteristics of *Psy* RAYR-BL compared to the genome reference *Pst* DC3000. For *Psy* RAYR-BL all characteristics are based on contigs ≥ 200 bp.

Feature	*Psy* RAYR-BL	*Pst* DC3000 ^a^		
Molecule	Draft genome	Chromosome	pDC300A	pDC3000B
Size	5,871,397	6,397,126	73,661	67,473
G+C content (%)	58.98	58.4	55.1	56.1
Number of contigs (≥ 200)	110	1	1	1
Largest contig	1,064,144	6,397,126	73,661	67,473
N50	193,008	6,397,126	73,661	67,473
L50	8	1	1	1
Genes	5188	5765	77	77
CDSs (coding)	5053	5466	74	68
rRNA	6	16	-	-
tRNA	57	63	-	-
Number of CDSs with assigned function	4453 (88.1%)	4556 (83.4%)	54 (72.9%)	47 (69.1%)
Number of CDSs without assigned function	600 (11.9%)	910 (16.6%)	20 (27.0%)	21 (30.9%)

^a^ The information about the *Pst* DC3000 assembly was obtained from NCBI (available at https://www.ncbi.nlm.nih.gov/assembly/GCF_000007805.1#/def, accessed on 11 November 2021).

**Table 2 microorganisms-10-00707-t002:** Identification of genes coding for enzymes of Phytotoxin biosynthetic pathways.

Phytotoxin	Reference Organism	Genes ID		Number of Genes in *Psy* RAYR-BL	
		Reference Genes	*Psy* RAYR-BL	Total	Percentage (%)
Coronatine	*Pseudomonas syringae* pv *tomato* DC3000	*cfl, cfa1, cfa2, cfa3, cfa4, cfa5, cfa6, cfa7, cfa8, cfa9, corR, corS, cmaD, cmaE, cmaA, cmaB, cmaC, cmaT*	-	-	-
Mangotoxin	*Pseudomonas syringae* pv. *syringae* UMAF0158	*mgoD, mgoA, mgoC, mgoB*, ORF2, ORF1	K0038_05071, K0038_05070,K0038_05069, K0038_05068, K0038_05067, K0038_05066,	6/6	100
Phaseolotoxin	*Pseudomonas savatanoi* pv *phaseolicola* 1448A	*argK, phtA, phtB, phtC, phtD, phtE, phtF, phtG, phtH, desO, phtJ, phtK, phtL, phtM, phtN, phtP, amtA, phtQ, phtS, phtT, phtU, phtV*	K0038_02673, K0038_02672, K0038_02670, K0038_02671, K0038_02666, K0038_02665, K0038_02664, K0038_02663, K0038_02662, K0038_02661, K0038_02660, K0038_02659, K0038_02668	13/22	59.1
Syringomycin and syringopeptin	*Pseudomonas syringae* pv *syringae* B728a	YP_235685.1, YP_235686.1, YP_235687.1, YP_235688.1, YP_235689.1, YP_235690.1, YP_235691.1, YP_235692.1, YP_235693.1,	K0038_02743, K0038_02745, K0038_02746, K0038_02747, K0038_02748, K0038_02749, K0038_02756, K0038_02757, K0038_02758	9/9	100

## Data Availability

Not applicable.

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
