# Peer review of "Molecular and Genomic Characterization of the Pseudomonas syringae Phylogroup 4: An Emerging Pathogen of Arabidopsis thaliana and Nicotiana benthamiana"

_microorganisms, 2022, doi:10.3390/microorganisms10040707_

Round 1

Reviewer 1 Report

This manuscript characterizes genomic and virulence traits of Pseudomonas syringae RAYR-BL, a highly virulent pathogen from Arabidopsis. The authors compare Pss RAYR-BL with DC3000, a well-known model bacterium, and found HopBJ1, a putative T3 effector absence from DC3000, involved in the virulence of Pss RAYR-BL. Although similar mechanisms have been described in other Pseudomonas species, this study provides initial steps for a better understanding of Pss RAYR-BL.

We have a few comments:

Remove L44-52.

L289, change one hundred twenty-nine to 129

Fig 6B, I wonder why rpoD is missing for Psy in planta?

Fig 8. Why use GFP-fused proteins?

Reviewer 2 Report

This is a very interesting study that combines genomic/bioinformatics analyses with wet lab experiments. The manuscript was well written and easy to follow. There is plenty of extra data in the supplementary material that may be useful to many experts in the field.

Major corrections:

It would be nice to summarize the statistics of paragraph 277-302, concerning the common and unique genes, in a table.

Line 311-314: The authors should mention here that another core proteome study of P. syringae strains that was based on 34 complete genomes/chromosomes identified 2944 core proteins and the protein count of the strains of this group ranged between 4973–6026 (average: 5465) (https://doi.org/10.3390/d12080289). The difference in core proteins between the two studies is related to the selection of the available genomes and on whether they comprise complete genomes or partially assembled one (contigs/scaffolds – that may be missing some genes – that could lower the number of core proteins of the group).

Figure 4: It would be much more informative if the phylogenomic tree was combined with a matrix to its right, that would show the presence and absence of the various genes of interest (discussed throughout the paper) in the various P. syringae strains. The authors could use Treedyn (doi: 10.1186/1471-2105-7-439.) or iTOL (doi: 10.1093/nar/gkab301.) for that.

Minor corrections:

Line 44-52: Please remove them.

Line 55-56: please rephrase.

Line 58: extreme

Line 86: assembly

Line 90: please rephrase.

Line 146: I am not sure I understand what this >10 kb means.

Line 155: I think its more accurate to say that the authors performed phylogenomic analyses.

In section 2.4, it would be nice to also mention here how many orthologous gene groups were used for the phylogenomic analysis and how many total amino acid collumns did the concatenated multiple alignment contain, after GBlocks filtering.

Line 176: identify

Line 178: syringomycin (and elsewhere).

Line 194: agroinfiltration?

Line 213: strain twice

Line 217: please leave space after accessions.

Line 238: infiltrated

Line 244: there was no statistical difference

Line 256: 6.54 Mbp.

Line 258: have an

Line 262: are displayed in Figure 2.

Line 288-289: I am a bit confused here. Shouldn't the orthogroup numbers (shared) be the same in the two strains?

Line 311: A phylogenomic tree was constructed…

Line 329: that participate in

Line 410: please rephrase.

Line 500: better change the term “producing”, maybe use “inducing”.
